# Low-Complexity Filter for Software-Defined Radio by Modulated Interpolated Coefficient Decimated Filter in a Hybrid Farrow

**DOI:** 10.3390/s22031164

**Published:** 2022-02-03

**Authors:** Temidayo O. Otunniyi, Hermanus C. Myburgh

**Affiliations:** Department of Electrical, Electronic and Computer Engineering, University of Pretoria, Pretoria 0002, South Africa; herman.myburgh@up.ac.za

**Keywords:** channelisation, Farrow filter, frequency response masking filter, fractional delay filter, coefficient decimation

## Abstract

Realising a low-complexity Farrow channelisation algorithm for multi-standard receivers in software-defined radio is a challenging task. A Farrow filter operates best at low frequencies while its performance degrades towards the Nyquist region. This makes wideband channelisation in software-defined radio a challenging task with high computational complexity. In this paper, a hybrid Farrow algorithm that combines a modulated Farrow filter with a frequency response interpolated coefficient decimated masking filter is proposed for the design of a novel filter with low computational complexity. A design example shows that the HFarrow filter bank achieved multiplier reduction of 50%, 70% and 64%, respectively, in comparison with non-uniform modulated discrete Fourier transform (NU MDFT FB), coefficient decimated filter bank (CD FB) and interpolated coefficient decimated (ICDM) filter algorithms. The HFarrow filter bank is able to provide the same number of sub-band channels as other algorithms such as non-uniform modulated discrete Fourier transform (NU MDFT FB), coefficient decimated filter bank (CD FB) and interpolated coefficient decimated (ICDM) filter algorithms, but with less computational complexity.

## 1. Introduction

Recent rises in new and emerging technologies warrant low-complexity channelisation algorithms for multi-standard software-defined radio (SDR) channels. Channelisation is one of the applications of software-defined radio (SDR) for processing channels of choice from wideband input channels and it is useful in signal processing [1] and image compression [2]. It usually takes place at the digital front-end of the receiver. The extent of computational complexity differs in different channelisation algorithms, from uniform channelisation algorithms, such as the per-channel (PC), pipelined/binary algorithm and pipelined frequency transform (PFT), to the non-uniform algorithms [3]. A Farrow filter bank is typically used for extracting uniform as well as non-uniform channels. However, at higher frequencies, its performance degrades and high computational complexities are required to extract such channels. The resultant effects of bigger filter length, longer filter coefficient, and the huge number of multipliers consumed by the Farrow filter, render it unfit for the upcoming mobile communication services. Therefore, the Farrow filter bank must be improved upon with low-complexity features.

A Farrow filter is a variable digital filter with adjustable control parameters. These control parameters may be adjustable with arbitrary delay. A Farrow filter is used in multi-standard receiver channelisation algorithms and for sample rate conversion [4]. It performs best at low frequencies but with significant performance degradation at higher frequency bands. The magnitude response of the Farrow structure is flat at low frequencies only, which limits its adaptability to other frequency bands. However, introducing an adaptive rational sample rate converter (SRC) and filtering at every branch of the channeliser allow different sub-band signals to be isolated, but with signal distortion.

Different attempts to reduce complexity and improve the performance of channelisation algorithms have been developed. A modified Farrow structure [5] has proven efficient for implementing rational sampling rate conversion by reducing the number of operators in the Farrow structure. A transposed modified Farrow structure [6,7] was proposed for reducing complexity in a Farrow filter by transposing the modified filter and thereby reducing the number of operators. Using a transposed modified Farrow structure, major advantages were lower sample conversion and reduced computational complexity [6,7]. Apart from these approaches, a variable cut-off frequency (VCF) method was developed to control the cut-off frequency of a variable filter in order to reduce its computational load [8]. A fixed-coefficient VCF filter with discrete control over frequency uses techniques such as a coefficient decimation method (CDM)-based filter and a frequency-response-masking-based filter. The coefficient decimation method (CDM) [9,10,11,12] was proposed to achieve low complexity by using one prototype filter or modal filter. Different signals with different passband widths and passband locations can be extracted with minimum overhead. Coefficient decimation method 1 (CDM-1) allows the extraction of different multiband frequency responses whereas CDM-11offers flexible passband width and passband centre frequency of the prototype filter [9,13,14,15]. A modified form of CDM known as the modified coefficient decimation method (MCDM) was proposed to offer a solution to the variable digital filter [16]. The MCDM technique provides higher frequency response flexibility and offers twice the centre frequency resolution than the classical coefficient decimation method (CDM). Two coefficient decimation operations can be performed on MCDM. One of the operations is for extracting different multi-band frequency responses (termed MCDM-1) and the other operation offers flexible passband width and passband centre frequency of the prototype filter (termed MCDM-11). The combination of coefficient decimation method 1 (CDM-1) and the modified coefficient decimation method (MCDM-1) is referred to as an improved coefficient decimation method (ICDM-1) [17], whereas the combination of coefficient decimation method 11 (CDM1-11) and modified coefficient decimation method 11 (MCDM-11) is termed an improved coefficient decimation method (ICDM-11). The main drawback of coefficient decimation is that the passband ripples and stop-band attenuation deteriorates in the frequency responses by a factor of *D*. The number of multipliers consumed by the CDM method [18] was 901.

A low-complexity, sharp-variable cut-off filter can be designed by using frequency response masking (FRM) [19]. Frequency response masking is used to extract multiple non-uniform channels. Hybridising a variable digital filter with FRM can result in lower computational complexity. FRM in combination with the CDM method was used for lowering the complexity of a variable filter [17,20]. Multi-stage FRM can also be used to reduce the complexity of a variable filter [21,22]. Weighted FRM can also be used for reducing the complexity of a variable filter [23,24]. Farrow interpolation [25,26] was designed for converting the sample rate by a fractional or rational factor. In this method, the $D^{\mathrm{th}}$ coefficients of the prototype filters are grouped together while discarding the remaining coefficients. L−1 zeros are inserted in between the selected coefficients. The number of multipliers used were very high [19]. An FRM-based tree structure non-uniform filter bank was designed to reduce the complexity of the filter to the minimum [27]. A substantial reduction in the complexity of the variable digital filter can be achieved by combining a modulated digital filter bank with the FRM approach, as can be seen with methods such as the non-uniform modulated discrete Fourier transform (MDFT) and modulated cosine filter [9]. A modulated generalised discrete Fourier transform (GDFT) filter bank was proposed to reduce the complexity of the Farrow filter [28]. The filter operates on a series of frequency-shifted oversampled sub-band signals. The approach attained −55 dB in reconstruction error.

Further complexity reductions were seen when frequency-response masking (FRM) and a non-maximally decimated filter bank (NMDUFB) were used [9]. A hybrid modified improved coefficient decimation method (HMICDM) was developed to improve the complexity of a variable digital filter [29]. The Lagrange interpolation method [30] was used for fractional delay approximation in a Farrow filter. The method increases the sampling rate of signals while the computational complexities are reduced to minimal. The lower complexity implementation and lower cost of a cosine modulated filter bank [3] can also be exploited.

Table 1 recaps the comparison study of the different channelisation algorithms.

From Table 1, it can be seen that computation complexity is still high for Farrow-based filters. The goal of this work is towards improving on this performance. The solution approach explored here involves the development of a modulated Farrow filter and the hybridisation of the modulated Farrow filter with a frequency response interpolated coefficient decimated masking filter. The filter based on the algorithm is simulated and tested using Matlab. It is hoped that the hybrid algorithm will improve the filter order, number of multipliers consumed, stop-band attenuation and pass band ripples, and gives a Farrow filter with low computational complexity.

The main novelties of this work are:Development of a low Farrow filter using a first-order differential method;Development of symmetrical frequency responses;Development of a modulation Farrow filter;Hybridisation of a modulation Farrow with frequency response masking filter and interpolation coefficient decimation filter.

## 2. Hybrid Farrow Filter Derivation

The algorithm development here involves modulation of a Farrow filter and its optimisation by using a hybrid of a frequency response masking (FRM)-based interpolated filter bank and a coefficient decimating filter (CD-1). The algorithm development is divided into two stages. The first stage involves the development of a low Farrow coefficient using a first-order differential method, development of symmetrical frequency responses and development of a modulating Farrow filter in which the first two steps above are used during its implementation. The second stage combines the modulating Farrow filter with a frequency masking filter and interpolation coefficient decimation to produce the algorithm referred to as the hybrid Farrow (HFarrow) filter algorithm in this context forthwith. The chronological order of the methods used is highlighted below.

Development of low Farrow filter using first-order differential method;Development of symmetrical frequency responses;Development of modulation Farrow filter;Hybridisation of modulation Farrow with frequency response masking filter and interpolation coefficient decimation filter.

Figure 1 shows the flowchart description of the investigation procedure for the HFarrow algorithm.

### 2.1. Farrow Interpolation Using First-Order Differential Approach

The Farrow structure was implemented using the LaGrange polynomial. It is a piecewise approximation of the filter into a polynomial form that shares a common set of coefficients, which results in the interpolation of input signals. Two important design parameters are polynomial order, *k*, and Farrow sub-filter, *N* [6,7,33]. It is implemented as a direct form of the FIR filter structure and it is obtained as an approximation of the continuous time function, $X_{c}\left(t\right)$, by fractional delay, *d*, as indicated in Equation (Equation 1) [34].
(1)$$\begin{array}{cc} y(n) & =h(d)*x(n)\\ y(n) & =h(n,d)*x(n)\\ \; & =\sum x\left(n\right)*C_{k}d^{k} \end{array}$$

The impulse response is computed using the Lagrange method. From the impulse response h(n,d), the fixed coefficients can be determined. The coefficients, $C_{k}$, from Equation (Equation 1) are derived from the set of N+1 linear equations. These coefficients are expressed in terms of fractional delay in such a way that 0≤d≤1. The filter coefficient, h(n), can be expressed in terms of $C_{k}$ as $C_{0}$ + $C_{1}$ + $C_{2}$ + … + $C_{n}$. The Farrow filter relies on a filter bank structure whereby each filter coefficient is approximated as the $N^{\mathrm{th}}$ order polynomial, *d*, as shown in Equation (Equation 2) [35].
(2)$$\begin{array}{c} h\left(n,d\right)=\sum c_{k}\left(n\right)d^{k}\\ n=0,1,\ldots ,N\\ 0\leq d\leq 1 \end{array}$$

Expressing Equation (Equation 2) in the *z*-domain, the filter transfer function is represented as in Equation (Equation 3).
(3)$$\begin{array}{c} \begin{array}{cc} H_{d}\left(z\right) & =\sum_{n=0}^{N}h\left(n,d\right)z^{-n}\\ \; & =\sum_{n=0}^{N}|\sum_{k=0}^{p}C_{k}\left(n\right)z^{-n}|d^{k}\\ \; & =|\sum_{k=0}^{p}C_{k}\left(z\right)d^{k}| \end{array} \end{array}$$
where $C_{k}\left(z\right)$ represents the set of M + 1 FIR sub-filters. From the relation in Equation (Equation 3), the filter structure is made up of a bank of fixed-weighted fractional delay, *d*, and summed u at the output of every tap.
(4)$$\begin{array}{cc} h(n,d) & =\prod_{\left(k=0,k\neq 0\right)}^{n}\frac{d-k}{n-k}\\ \; & ={\left(-1\right)}^{\left(N-n\right)}\left(\begin{array}{c} d\\ n \end{array}\right)\left(\begin{array}{c} d-n-1\\ N-n \end{array}\right)\\ \; & =\frac{d}{n}X\frac{d-1}{n-1}X\frac{d-n+1}{1}X\frac{d-n-1}{-1}X\frac{d-n}{n-N}\\ \mathrm{for}\;n & =\;0,1,2,3,\ldots ,N \end{array}$$

When N=3 and the fractional delay is *d*, the impulse response is shown in Equations (Equation 5) and (Equation 7).
(5)$$\begin{array}{c} h\left(n,d\right)=\prod_{\left(k=0,k\neq 0\right)}^{3}\frac{d-k}{n-k}\\ \mathrm{for}\;n=0,1,2,3 \end{array}$$

The coefficient for the fourth-order poly-phase filter is calculated using Equations (Equation 6), (Equation 7), (Equation 9) and (Equation 8).
(6)$$\begin{array}{cc} h(0,d) & =\prod_{\left(k=0,k=1,k\neq 0\right)}^{3}\frac{d-k}{0-k}\\ \; & =\frac{d-1}{-1}\times \frac{d-2}{-2}\times \frac{d-3}{-3}\\ \; & =\frac{1}{6}\left(d^{3}-6d^{2}-8d-6\right)\\ h(1,d) & =\prod_{\left(k=0,k=2,k\neq 1\right)}^{3}\times \frac{d-k}{1-k}\\ \; & =\frac{d}{1}\times \frac{d-2}{-1}\times \frac{d-3}{-2}\\ \; & =\frac{1}{2}\left(d^{3}-5d^{2}+6d\right)\\ h(2,d) & =\prod_{\left(k=0,k=1,k\neq 2\right)}^{2}\frac{d-k}{2-k}\\ \; & =\frac{d}{2}\times \frac{d-1}{1}\times \frac{d-3}{-1}\\ \; & =\frac{-1}{2}\left(d^{3}-4d^{2}+3d\right)\\ h(3,d) & =\prod_{\left(k=0,k=1,k\neq 3\right)}^{3}\frac{d-k}{3-k}\\ \; & =\frac{d}{3}\times \frac{d-1}{-2}\times \frac{d-2}{1}\\ \; & =\frac{1}{6}\left(d^{3}-3d^{2}+2d\right) \end{array}$$
(7)$$\begin{array}{c} H_{d}\left(z\right)=\sum_{n=0}^{N}\left(h,d\right)z^{-n}=h\left(0,d\right)+h\left(1,d\right)z^{-1}+h\left(2,d\right)z^{-2}\\ =\frac{1}{6}\left(d^{3}-6d^{2}-8d-6\right)+\frac{1}{2}\left(d^{3}-5d^{2}+6d\right) \end{array}$$
(8)$$\begin{array}{cc} C_{0}\left(z\right) & =1\\ C_{1}\left(z\right) & =\frac{8}{6}+3z^{-1}-\frac{3}{2}z^{-2}+\frac{2}{6}z^{-3}\\ C_{2}\left(z\right) & =\frac{-5}{6}-\frac{5}{2}z^{-1}+2z^{2}-\frac{1}{2}z^{-3}\\ C_{3}\left(z\right) & =\frac{1}{6}+\frac{1}{2}z^{-1}-\frac{1}{2}z^{-2}+\frac{1}{6}z^{-3}\\ \bar{C(z)} & =\Phi^{T}\bar{z}=\left[\begin{array}{c} C_{0}\left(z\right)\\ C_{1}\left(z\right)\\ C_{2}\left(z\right)\\ C_{3}\left(z\right) \end{array}\right] \end{array}$$
(9)$$\Phi =\left[\begin{array}{cccc} 1 & 0 & 0 & 0\\ \frac{-8}{6} & 3 & \frac{-3}{2} & \frac{-1}{3}\\ \frac{6}{6} & \frac{-5}{2} & 2 & \frac{-1}{2}\\ \frac{-1}{6} & \frac{1}{2} & \frac{-1}{2} & \frac{1}{6} \end{array}\right]$$

The number of operators is further reduced by finding the first derivatives of each filter impulse, h(n,d), as shown in Equation (Equation 10).
(10)$$\begin{array}{c} \begin{array}{cc} h^{\prime }\left(0,d\right) & =\frac{1}{6}\left(3d^{2}-12d-8\right)\\ h^{\prime }\left(1,d\right) & =\frac{1}{2}\left(3d^{2}-10d+6\right)\\ h^{\prime }\left(2,d\right) & =\frac{-1}{2}\left(3d^{2}-8d+3\right)\\ h^{\prime }\left(3,d\right) & =\frac{1}{6}\left(3d^{2}-6d+2\right) \end{array} \end{array}$$
(11)$$\begin{array}{cc} C_{0}\left(z\right) & =\frac{8}{6}+3z^{-1}-\frac{3}{2}z^{-2}+\frac{1}{3}z^{-3}\\ C_{1}\left(z\right) & =-2-5z^{-1}+4z^{-2}-z^{-3}\\ C_{2}\left(z\right) & =\frac{1}{2}-\frac{3}{2}z^{-2}+\frac{1}{2}z^{-3}\\ C_{3}\left(z\right) & =0+0-0+0\\ \bar{C(z)} & =\Phi^{T}\bar{z}=\left[\begin{array}{c} C_{0}\left(z\right)\\ C_{1}\left(z\right)\\ C_{2}\left(z\right)\\ C_{3}\left(z\right) \end{array}\right] \end{array}$$
(12)$$\Phi =\left[\begin{array}{cccc} \frac{8}{6} & 3 & \frac{3}{2} & \frac{1}{3}\\ 2 & -5 & 4 & -1\\ \frac{1}{2} & \frac{3}{2} & \frac{-3}{2} & \frac{1}{2}\\ 0 & 0 & 0 & 0 \end{array}\right]$$

Figure 2 shows the Farrow sub-filters. The filter is made up of fixed filters weighted by the fractional delay *d* and summed at the output of each tap. Figure 2 shows shared elements such as unit delays, which make the structure very efficient.

### 2.2. Exploring Frequency Responses of Symmetrical Farrow Filter Polynomial Functions

The continuous impulse response, $h_{c}\left(t\right)$, of the Farrow structure is represented as a linear combination of basis functions as can be seen in Equation (Equation 13) [34,36].
(13)$$h_{c}\left(t\right)=\left\{\begin{array}{cc} \sum_{m=0}^{M}C_{k}\left(n\right){\left(\frac{t}{T_{i}}+t_{n}\right)}^{n}, & \left(d_{min}-t_{n}\right)T_{1}\leq t\leq \left(d_{max}-t_{n}\right)T_{k}\\ \; & \;\mathrm{for}\;n\;=\;0,1,2,\;\ldots \ldots .,\;N\;\\ 0, & otherwise \end{array}\right.$$
where $t_{n}$ is the base-point value, $T_{1}$ is the input sampling frequency and $C_{k}$ is the $k^{\mathrm{th}}$ sub-filter coefficient. The interval of the fractional delay d is represented as follows in Equation (Equation 14).
(14)$$\begin{array}{c} \frac{N-1}{2}\leq d\leq \frac{N+1}{2}\\ n=\left\lfloor \frac{t}{T_{1}}\right\rfloor \\ d=\frac{t_{1}}{T_{1}}-d \end{array}$$

The inter-sample interval is given as in Equation (Equation 15).
(15)0≤d≤1

By introducing the basis function for piecewise polynomials that are zero outside a given interval, Equation (Equation 16) was derived.
(16)$$f\left(m,d\right)=\left\{\begin{array}{cc} d^{k}, & d_{min}\leq d\leq d_{max}\\ 0, & otherwise \end{array}\right.$$

Therefore, the continuous impulse response, $h_{c}\left(t\right)$, is shown in Equation (Equation 17).
(17)$$h_{c}\left(t\right)=\sum_{n=0}^{N}\sum_{m=0}^{M}C_{k}f\left(m,d\right)$$

The frequency domain representation of the impulse response is represented in Equation (Equation 18).
(18)$$\begin{array}{cc} H(e^{jw},d) & =\sum_{n=0}^{M}C_{k}\left(\omega \right)d^{k}\\ H(\omega ,d) & =\sum_{m=0}^{M}C_{k}\left(\omega \right)d^{k}\\ \; & =\sum_{m=0}^{M}C_{k}\left(\omega \right)G\left(\omega \right)\\ \;\mathrm{where}\;G(\omega ) & =d^{k} \end{array}$$

Then, by applying a Fourier transform, the fractional delay G(ω) can be expressed as indicated in Equation (Equation 19).
(19)G(k,m,ω)=F{g(m,n,ω)}

The frequency variable ω is normalised to the input sampling period, that is, $\omega =\Omega T_{1}$, and delay *d* is replaced with μ. In order to obtain the continuous frequency responses, three cases are considered. The first to consider is when *N* is odd while the second one for consideration is when *N* is even and finally when N>0.
(20)$$G\left(m,n,\omega \right)=\int_{-\infty }^{\infty }g\left(m,n,\omega \right)e^{-j\omega \mu }d\mu$$

For odd N,
(21)$$g\left(m,n,\omega \right)=\left\{\begin{array}{cc} {\left(\mu +n+\frac{1}{2}\right)}^{k}, & -n-1\leq \mu \leq -n\\ {\left(-1\right)}^{k}{\left(\mu -n-\frac{1}{2}\right)}^{k}, & n\leq \mu \leq n+1\\ 0, & otherwise \end{array}\right.$$

Therefore, the discrete version of fractional delay can be expressed as shown in Equation (Equation 22).
(22)$$\begin{array}{cc} G(m,n,\omega ) & =\int_{-\infty }^{\infty }g\left(m,n,\omega \right)e^{-j\omega \mu }d\mu \\ \; & =\int_{-n-1}^{-n}{\left(\mu +n+\frac{1}{2}\right)}^{k}\omega e^{-j\omega \mu }d\mu +\int_{n}^{n+1}{\left(-1\right)}^{k}{\left(\mu -n-\frac{1}{2}\right)}^{k}\omega e^{-j\omega \mu }d\mu \\ \; & =\int_{-\frac{1}{2}}^{\frac{1}{2}}\mu_{1}\omega e^{-j\omega \mu_{1}-\left[n+\frac{1}{2}\right]}d\mu +\int_{-\frac{1}{2}}^{\frac{1}{2}}{\left(-1\right)}^{m}\mu_{2}\omega e^{-j\omega \mu_{2}-\left[n+\frac{1}{2}\right]}d\mu \\ \; & =e^{j(n+\frac{1}{2}\omega )}\int_{-\frac{1}{2}}^{\frac{1}{2}}\mu_{1}\omega e^{-j\omega \mu_{1}}d_{\mu 1}+{\left(-1\right)}^{m}e^{j(n+\frac{1}{2})}\omega \int_{-\frac{1}{2}}^{\frac{1}{2}}{\left(-1\right)}^{m}\mu_{2}\omega e^{-j\omega \mu_{2}}d_{\mu 2}\\ \; & =\left[e^{j(n+\frac{1}{2})\omega }+{\left(-1\right)}^{m}e^{j(n+\frac{1}{2})\omega }\right]\int_{-\frac{1}{2}}^{\frac{1}{2}}\mu_{1}\omega e^{-j\omega d_{\mu }} \end{array}$$

Fractional delay G(m,n,ω) can be expressed as a function of two variables. That is, ϕ(m,n,ω) and ψ(m,ω).
(23)G(m,n,ω)=ϕ(m,n,ω)ψ(m,ω)
(24)$$\mathrm{where}\;\phi \left(m,n,\omega \right)=\left\{\begin{array}{cc} 2cos(\left[n+\frac{1}{2}\right]\omega ), & \;\mathrm{for}\;m=\mathrm{even}\;,N=odd\\ 2jsin(\left[n+\frac{1}{2}\right]\omega ), & \;\mathrm{for}\;m=\mathrm{even}\;,N=odd \end{array}\right.$$

In another instance, when N=even and n>0, then
(25)$$\begin{array}{cc} G(m,n,\omega ) & =\int_{-n-\frac{1}{2}}^{-n+\frac{1}{2}}{\left(\mu +n\right)}^{k}\omega e^{-j\omega \mu }d\mu +\int_{n-\frac{1}{2}}^{n+\frac{1}{2}}{\left(-1\right)}^{k}{\left(\mu -n\right)}^{k}\omega e^{-j\omega \mu }d\mu \\ G(m,n,\omega ) & =\left[e^{j\omega }+{\left(-1\right)}^{m}e^{j\omega }\right]\int_{-\frac{1}{2}}^{\frac{1}{2}}\mu^{m}e^{-j\omega }\mu_{k}\\ \; & =\phi (m,n,\omega )\psi (m,\omega )\\ \mathrm{where}\;\psi (m,\omega ) & =\int_{-\frac{1}{2}}^{\frac{1}{2}}\mu^{m}e^{-j\omega }d_{\mu } \end{array}$$
(26)$$G\left(m,n,\omega \right)=\phi \left(m,n,\omega \right)=\left\{\begin{array}{cc} 2cos(n\omega ), & m=even,N=odd\\ 2jsin(n,\omega ), & m=odd,N>0 \end{array}\right.$$

Lastly, for n=even, n=0andm=even
(27)$$\begin{array}{cc} G(m,0,\omega ) & =\int_{-\frac{1}{2}}^{\frac{1}{2}}{\left(\mu \right)}^{k}\omega e^{-j\omega \mu }d\mu +\int_{n-\frac{1}{2}}^{n+\frac{1}{2}}{\left(-1\right)}^{k}{\left(\mu -n\right)}^{k}\omega e^{-j\omega \mu }d\mu \end{array}$$
(28)$$G\left(m,0,\omega \right)=\phi \left(m,n,\omega \right)\mathrm{with}=\left\{\begin{array}{cc} \phi (m,n,\omega )=1, & N=even,m=even,n=0\\ \phi (m,n,\omega )=0, & N=even,m=odd,n=0 \end{array}\right.$$

The different variants of the basis functions G(m,n,ω) are represented as follows in Equation (Equation 29).
(29)$$\phi \left(m,n,\omega \right)=\left\{\begin{array}{cc} 1, & N=even,n=0,m=even\\ 0, & N=even,n=0,m=odd\\ 2cos(n,\omega ), & N=even,n>0,m=even\\ 2jsin(n,\omega ), & N=even,n>0,m=odd\\ 2cos(\left[n+\frac{1}{2}\right],\omega ), & N=odd,m=even\\ 2jsin(\left[n+\frac{1}{2}\right],\omega ), & N=odd,m=even \end{array}\right.$$

Considering the scaling function as shown in Equation (Equation 30): (30)$$\psi \left(m,\omega \right)=\int_{-\frac{1}{2}}^{\frac{1}{2}}\mu^{m}e^{-j\omega \mu }d\mu$$

By multiplying the integral with a unit rectangle ∏(μ), Equation (Equation 31) is obtained.
(31)$$\psi \left(m,\omega \right)=\int_{-\frac{1}{2}}^{\frac{1}{2}}\prod \left(\mu \right)\mu^{m}e^{-j\omega \mu }d\mu$$

Expressing the scaling function in terms of a Fourier transform:(32)$$\begin{array}{cc} \psi (m,\omega ) & =F\{\prod \left(\mu \right)\mu^{m}\}\\ \; & =j^{m}\frac{\mu^{m}}{\mu \omega^{m}}sinc\left(\frac{\omega }{2}\right) \end{array}$$

The basis function G(m,n,ω) = ψ(m,ω)ϕ(m,n,ω) is both real and even. This implies that $H_{c}\left(j\omega \right)$ is both a real and even function and is a derivative of the real-valued and symmetrical nature of $h_{c}\left(t\right)$. The basis function G(m,n,ω) can be represented as real-valued functions for ϕ(m,n,ω) and ψ(m,ω) by transposing the imaginary unit *j* from ϕ(m,n,ω) to ψ(m,ω) for odd *m*. This can be represented as shown in Equation (Equation 33).
(33)$$G\left(m,n,\omega \right)=\psi \left(m,\omega \right)\phi \left(m,n,\omega \right)\;\mathrm{with}\;\psi \left(m,\omega \right)={\left(-1\right)}^{m}\frac{\mu^{m}}{\mu \omega^{m}}sinc\left(\frac{\omega }{2}\right)$$

Expressing the scaling function as a real value as indicated by Equation (Equation 34):(34)$$\begin{array}{cc} \psi (m,\omega ) & =sinc(\frac{\omega }{2})\\ \psi (m,\omega ) & =\sum_{k=0}^{\infty }a_{k}\omega^{k}\\ \mathrm{where}\;\sum_{k=0}^{\infty }a_{k}\omega^{k}\;\mathrm{with}\;a_{k} & =\left\{\begin{array}{cc} \frac{{\left(-1\right)}^{\frac{k+m}{2}}\left(k+1\right)}{2^{k+m}\left(k+m+1\right)!}, & k+m\;\mathrm{is}\;\mathrm{even}\;\\ 0, & k+m\;\mathrm{is}\;\mathrm{odd}\; \end{array}\right. \end{array}$$

The values for ψ(m,ω) are identical for the real variant of ψ(m,0) when ω=0 as indicated in Equation (Equation 34).
(35)$$\begin{array}{ccc} \psi (m,0) & =j^{m}a_{0}^{m} & =\left\{\begin{array}{cc} \frac{{\left(-1\right)}^{\frac{m}{2}}}{\left(m+1\right)!2^{m}}, & m\;\mathrm{is}\;\mathrm{even}\;\\ 0, & m\;\mathrm{is}\;\mathrm{odd}\; \end{array}\right. \end{array}$$

The impulse response is scaled proportionally and this is able to reduce the distortion or numerical oscillation of the impulse response to minimal.

### 2.3. Modulation of Farrow Filter

Having seen the symmetrically scaled impulse response of the Farrow filter, the Farrow filter can be modulated to reduce the computational complexity. The value of μ can be determined using Equation (Equation 36) with channel centre frequency $\omega_{k}$ and is represented as shown in Equation (Equation 37) [34].
(36)$$\frac{\pi }{\omega_{s}}$$
(37)$$\omega_{k}=\frac{2\pi (k+k_{0})}{K}$$
where $\omega_{k}$ is the centre frequency of the channels.

The modulated Farrow filter can be represented as shown in Equation (Equation 38).
(38)$$\begin{array}{cc} H(e^{jw},\mu ) & =\Sigma_{m=0}^{M}C_{k}\left(\omega \right)G\left(\omega \right)\\ \; & =\Sigma_{m=0}^{M}C_{k}\left(\omega \right)\left(2cos\left(n,\omega \right)+2jsin\left(n,\omega \right)\right)\\ \; & =\Sigma_{m=0}^{M}C_{k}\left(\omega \right)\left[e^{j\omega n\mu }+{\left(-1\right)}^{m}e^{j\omega n\mu }\right] \end{array}$$

Thus, by modulating Equation (Equation 1) with $e^{-j\omega md}$, newly generated Farrow filter $y^{\prime }\left(n\right)$ is obtained as shown in Equation (Equation 39).
(39)$$\begin{array}{cc} y(n) & =h(n,\mu )*x(n)\\ y(n) & =h(n,\mu )*x(n)\\ y^{\prime }\left(n\right) & =\sum x\left(n\right)*h\left(\omega ,\mu \right)e^{-j\omega Mn\mu } \end{array}$$

If the channel signal is critically sampled, the decimation ratio and the bandwidth are related as follows in Equation (Equation 40).
(40)$$M=\frac{K}{2}$$

Thus, the new Farrow filter bank can be reduced to the following filter bank in Equation (Equation 41).
(41)$$\begin{array}{cc} y_{k}^{\prime }\left(m\right) & ={\left(-1\right)}^{km}\Sigma_{n=-M-1}^{N}C_{k}\left(n\right)h_{k}\left(mK-1\right)x_{(}n) \end{array}$$

If *m* is replaced by *L* and *K* is replaced by 2M, and if linear-phase prototype low-pass filter H(z) of order *N* has a pass-band edge of $\theta_{a}=\frac{2m\pi -\omega_{\omega_{\Delta }}}{M}$ and stop-band edge of $\phi_{a}=\frac{2m\pi -\omega_{\omega_{\Delta }}}{M}$, with $\omega_{\Delta }$ as the width of the transition band, then for even multiples of the number M of sub-bands, the length is N+1; that is, N=(2LM−1).

Here, the impulse response $h_{k}\left(mK-1\right)$ will be reduced as follows. It can be seen that $h_{k}$ contains terms that multiply $h_{n}$ and this can be expressed as variable *D* as seen in Equation (Equation 42).
(42)$$\begin{array}{c} h_{k}\left(n\right)=C_{k}\left(n\right)h\left(n\right)(2cos\left\{\left(2M+1\right)\frac{\pi }{2M}\left[\left(N+2kM\right)-\frac{N}{2}+\Phi \right]\right\}\\ +2jsin\left\{\left(2M+1\right)\frac{\pi }{2M}\left[\left(N+2kM\right)-\frac{N}{2}+\Phi \right]\right\}) \end{array}$$
(43)$$\begin{array}{c} \;\mathrm{where}\;e^{j\omega n1d}=2cos\left\{\left(2M+1\right)\frac{\pi }{2M}\left[\left(N+2kM\right)-\frac{N}{2}+\Phi \right]\right\}\\ {\left(-1\right)}^{m}e^{j\omega n2d}=2jsin\left\{\left(2M+1\right)\frac{\pi }{2M}\left[\left(N+2kM\right)-\frac{N}{2}+\Phi \right]\right\} \end{array}$$

However, by exploiting symmetry in Equations (Equation 43) and (Equation 44) this becomes
(44)$$\begin{array}{c} h_{k}\left(n\right)=C_{k}\left(n\right)h\left(n\right)\left(2cos\left\{\left(2M+1\right)\frac{\pi }{2M}\left[\left(N+2kM\right)-\frac{N}{2}+\Phi \right]\right\}\right)\\ h_{k}\left(n\right)=C_{n}h\left(n\right)\left[e^{j\omega n1\mu }\right]\psi \left(m,\omega \right) \end{array}$$

The poly-phase representation of the analysis filter bank can be described as shown in Equation (Equation 45).
(45)$$\begin{array}{c} H_{k}\left(z\right)=\sum_{n=0}^{N}C_{k}\left(n\right)h_{k}\left(n\right)z^{\left(-n\right)}\\ H_{k}\left(z\right)=\sum_{l=0}^{L-1}\sum_{j=0}^{2M-1}\left[e^{j\omega n1\mu 1}\psi \left(m,\omega \right)+e^{j\omega n2\mu 2}\psi \left(m,\omega \right)\right]C_{k}\left(2LM+j\right)h_{k}\left(2LM+j\right)z^{-(2LM+j)} \end{array}$$

The prototype filter can be decomposed into 2M poly-phase components, as follows in Equation (Equation 46), where $S_{j}\left(z\right)=\sum_{i=0}^{L-1}C_{k}\left(2LM+j\right)h_{k}\left(2LM+j\right)z^{\left(-L\right)}$ are the poly-phase components of the filter H(z).
(46)$$\begin{array}{cc} H(z) & =\sum_{j=0}^{2M-1}z^{-j}\sum_{l=0}^{L-1}\psi \left(m,\omega \right)\left[e^{j\omega n1\mu }+e^{j\omega n2\mu }\right]C_{k}\left(2LM+j\right)h_{k}\left(2LM+j\right)z^{\left(-2M-j\right)}\\ \; & =\sum_{j=0}^{2M-1}z^{-j}\sum_{l=0}^{L-1}\psi \left(m,\omega \right)\left[e^{j\omega n1\mu }+{\left(-1\right)}^{m}e^{j\omega n2\mu }\right]C_{k}\left(2LM+j\right)h_{k}\left(2LM+j\right)z^{\left(-2M-j\right)}\\ \; & =\sum_{j=0}^{2M-1}\psi \left(m,\omega \right)\left[e^{j\omega n1\mu }+{\left(-1\right)}^{m}e^{j\omega n2\mu }\right]z^{-j}S_{j}\left(z^{-2M}\right) \end{array}$$

Representing $D_{1}$= $\psi \left(m,\omega \right)\left(e^{j\omega n1\mu 1}\right)$ and $D_{2}$= $\psi \left(m,\omega \right)\left(e^{j\omega n2\mu 1}\right)$.

From Equation (Equation 47), it can be shown that $D_{1}$ and $D_{2}$ are M×N matrices, whose (m,j) elements are $D_{m,j}$ and $D_{m,j+m}$, respectively, for m,j=0,1,⋯(m−1).
(47)$$H\left(z\right)=\left[\begin{array}{c} H_{0}\left(z\right)\\ H_{1}\left(z\right)\\ \vdots \\ \vdots \\ H_{m-1}\left(z\right) \end{array}\right]=\left[\begin{array}{cc} D_{1} & D_{2} \end{array}\right]\left[\begin{array}{c} S_{0}\left(z^{2M}\right)\\ z^{-1}S_{1}\left(z^{2M}\right)\\ \vdots \\ \vdots \;\\ z^{\left(-2m-1\right)}S_{2m-1}\left(z^{2M}\right) \end{array}\right]$$

Representing δ(z) as shown in Equation (Equation 48):(48)$$\begin{array}{cc} \delta (z) & ={\left[\begin{array}{cccc} 1 & z^{-1} & \cdots & z^{-M+1} \end{array}\right]}^{T} \end{array}$$

The poly-phase representation of the matrix can be represented as in Equation (Equation 50), where S(z) is the poly-phase matrix.
(49)$$H\left(z\right)=\left[\begin{array}{cc} D_{1} & D_{2} \end{array}\right]\left[\begin{array}{ccc} S_{0}\left(z^{2M}\right) & 0 & \\ \; & S_{1}\left(z^{2M}\right) & \\ \; & \ddots & \\ \; & \;\ddots \;\\ \; & 0 & S_{2m-1}\left(z^{2M}\right) \end{array}\right]\left[\begin{array}{c} \delta (z)\\ z^{-m}\delta \left(z\right) \end{array}\right]$$
(50)$$H\left(z\right)=\left\{\begin{array}{c} D_{1}\left[\begin{array}{ccc} S_{0}\left(z^{2M}\right) & 0 & \\ \; & S_{1}\left(z^{2M}\right) & \\ \; & \ddots & \\ \; & \;\ddots \;\\ \; & 0 & S_{2m-1}\left(z^{2M}\right) \end{array}\right]\\ +\\ z^{-m}D_{2}\left[\begin{array}{ccc} S_{0}\left(z^{2M}\right) & 0 & \\ \; & S_{M+1}\left(z^{2M}\right) & \\ \; & \ddots & \\ \; & \;\ddots \;\\ \; & 0 & S_{2m-1}\left(z^{2M}\right) \end{array}\right] \end{array}\right\}\delta \left(z\right)$$

### 2.4. Combined Modulated Farrow and Interpolated Coefficient Decimated Filter

After improvements obtained from the modulation algorithm, further work was performed to achieve some improvements by hybridising the modulated algorithm with the frequency response masking algorithm [15].

The design involves a hybrid of a frequency response masking (FRM)-based interpolated coefficient decimated filter and a modulated Farrow filter.

The hybrid Farrow (HFarrow)-based filter bank consists of two branches; namely, the upper and the lower branches. The upper branch is made up of the FRM coefficient decimated filter and the masking filter, whereas the lower branch consists of the complementary FRM coefficient decimated filter and the complementary masking filter. A low-pass coefficient base interpolated linear phase FIR filter, $H_{a}\left(z^{\frac{L}{M}}\right)$, is formed from the cascade of the base interpolating filter, $H_{a}\left(z^{L}\right)$, and the coefficient decimating filter, $H_{cd}\left(z^{L}\right)$, to extract the sharp narrow-band channel of choice.

In addition, a bandpass edge complementary coefficient base interpolating filter, $H_{c}\left(z^{\frac{L}{M}}\right)$, is formed from the cascade of the complementary base interpolating filter, $H_{a}^{\prime }\left(z^{L}\right)$, and the complementary coefficient decimating filter, $H_{cd}^{\prime }\left(z^{M}\right)$, to isolate multi-band frequency responses. The low-pass coefficient base interpolated filter, $H_{a}\left(z^{L/M}\right)$, cascades with the farrow masking filter, $A_{k}\left(z\right)$, in the upper branch while the bandpass complementary coefficient base interpolating filter, $H_{c}\left(z^{L/M}\right)$, cascades with the complementary masking filter, $B_{k}\left(z\right)$, in the lower branch to produce low computational multi-narrow frequency bands.

The transfer function of the FRM coefficient decimation filter is given using Equation (Equation 51).
(51)$$\begin{array}{cc} H(z) & =\frac{L}{M}\left[H_{a}\left(z^{\frac{L}{M}}\right)A\left(z\right)+H_{c}\left(z^{\frac{L}{M}}\right)B\left(z\right)\right] \end{array}$$

The coefficient decimated base and complementary filters are symmetrical and asymmetrical linear phase FIR filter which can be expressed as $H_{a}\left(z^{\frac{L}{M}}\right)=H_{c}\left(-z^{\frac{L}{M}}\right)$. A half-band filter is introduced into the coefficient decimated based FRM filter to further reduce its computation complexity. This is possible as a result of the symmetrical properties possessed by the half-band filter. The time-domain impulse response of the CD-1 technique requires every other component to be zero except the components at the centre. That indicates that it is symmetrical around the centre. This translates to reduced complexity in terms of the number of the multiplies required by the filter.

The transfer function of the half-band masking FRM coefficient decimated filter band can be expressed in terms of two polyphase components as shown in Equation (Equation 52).
(52)$$\begin{array}{cc} H_{a}\left(z^{\frac{L}{M}}\right) & =\frac{L}{M}\left[H_{a0}\left(z^{\frac{2L}{M}}\right)+z^{-\frac{2L}{M}}H_{a1}\left(z^{\frac{2L}{M}}\right)\right]\\ H_{c}\left(z^{\frac{L}{M}}\right) & =\frac{L}{M}\left[H_{a0}\left(z^{\frac{2L}{M}}\right)-z^{-\frac{2L}{M}}\frac{1}{M}H_{a1}\left(z^{\frac{2L}{M}}\right)\right] \end{array}$$

The masking filters are replaced with two farrow filters as shown in the Figure 3. Masking filters $A_{k}\left(z\right)$ and $B_{k}\left(z\right)$ extract one or several pass-bands of the periodic model filter $H_{a}\left(z^{\frac{L}{M}}\right)$ and the complementary periodic model filter $H_{c}\left(-z^{\frac{L}{M}}\right)$. The transfer function for the HFarrow masking algorithm can be expressed as shown in Equation (Equation 53).
(53)$$\begin{array}{cc} H_{a}\left(z^{\frac{2L}{M}}\right) & =\frac{L}{M}\left[\left(H_{a0}\left(z^{\frac{2L}{M}}\right)+z^{-\frac{2L}{M}}H_{a1}\left(z^{\frac{2L}{M}}\right)\right)A\left(z\right)\right] \end{array}$$

The impulse response of the HFarrow channelisation algorithm is approximated with different fixed delay variables using polynomial interpolation methods. A set of sub-filters with fixed delay $\mu_{i}$, such that i=0,1,⋯i−1, is designed. The impulse response is interpolated by the $L^{\mathrm{th}}$ order using μ as the variable. The resultant impulse response is the interpolated version of HFarrow parameterised by delay μ. The parameter μ allows full unabridged control over the available bandwidth and the cut-off frequencies of the multi-band channels.

The input band of the masking FRM-CD signal is decomposed into multiple sub-bands, each with distinct bandwidth (BW). Each bandwidth is phase-shift modulated by fractional delay $\mu^{k}$. Applying the poly-phase decomposition as seen in Equation (Equation 46) will result in Equation (Equation 54), where $S_{ki}\left(z^{-2M}\right)$ are the *K* poly-phase components of $A_{k}\left(z\right)$ and $B_{k}\left(z\right)$.
(54)$$\begin{array}{cc} A_{k}\left(z\right) & =\sum_{n=0}^{2M-1}\psi \left(m,\omega \right)\left(e^{j\omega n1\mu }+{\left(-1\right)}^{m}e^{j\omega n2\mu }\right)z^{-n}S_{n}\left(z^{-2M}\right)\\ B_{k}\left(z\right) & =\sum_{n=0}^{2M-1}\psi \left(m,\omega \right)\left(e^{j\omega n1\mu }-{\left(-1\right)}^{m}e^{j\omega n2\mu }\right)z^{-n}S_{n}\left(z^{-2M}\right) \end{array}$$

Finally, Equation (Equation 55) shows the representation of each of the modulated masking bandpass filters and the block diagram for the hybrid Farrow masking filter is depicted in Figure 4.
(55)$$\begin{array}{cc} H_{a}\left(z^{\frac{2L}{M}}\right) & =\frac{L}{M}\left[H_{a0}\left(z^{\frac{2L}{M}}\right)+z^{\frac{-2L}{M}}H_{a1}\left(z^{\frac{2L}{M}}\right)\sum_{n=0}^{2M-1}\psi \left(m,\omega \right)\left(e^{j\omega n1\mu }+{\left(-1\right)}^{m}e^{j\omega n2\mu }\right)z^{-n}S_{n}\left(z^{-\frac{2L}{M}}\right)\right]\\ H_{a}\left(z^{\frac{2L}{M}}\right) & =\frac{L}{M}\left[H_{a0}\left(z^{\frac{2L}{M}}\right)+z^{-\frac{2L}{M}}H_{a1}\left(z^{\frac{2L}{M}}\right)\sum_{n=0}^{2M-1}z^{-n}\left(D_{1}+D_{2}\right)S_{n}\left(z^{-\frac{L}{M}}\right)\right] \end{array}$$

The transfer function of the FRM coefficient decimation filter is given using Equation (Equation 56).
(56)$$\begin{array}{cc} H(z) & =\frac{L}{M}\left[H_{a}\left(z^{\frac{L}{M}}\right)A\left(z\right)+H_{c}\left(z^{\frac{L}{M}}\right)B\left(z\right)\right]\\ \;\mathrm{But},\;\\ H_{a}\left(z^{\frac{L}{M}}\right) & =H_{c}\left(-z^{\frac{L}{M}}\right)\\ \;\mathrm{Therefore},\;\\ H(z^{\frac{2L}{M}}) & =\frac{2L}{M}\left[H_{a0}\left(z^{\frac{2L}{M}}\right)+z^{-\frac{2L}{M}}H_{a1}\left(z^{\frac{2L}{M}}\right)\sum_{n=0}^{2M-1}z^{-n}\left(D_{1}+D_{2}\right)S_{n}\left(z^{-\frac{2L}{M}}\right)\right] \end{array}$$

The final output sequence, y(n), can be expressed in terms of the convolution of x(n) and the filter transfer function, h(n).
(57)$$\begin{array}{c} y(n)=x(n)*h(n)\\ Y(z)=X(z)H(z) \end{array}$$

Figure 4 illustrates the HFarrow channelisation algorithm. Different signal sub-bands, $S_{n}\left(z^{-\frac{2L}{M}}\right)$, can be derived or obtained from masking filter $H_{a}\left(-z^{\frac{L}{M}}\right)$ and prototype complementary masking filter $H_{c}\left(-z^{\frac{L}{M}}\right),$ respectively.

The centre frequency component of each band is shifted precisely by a phase of fractional delay, μ. The phase to be shifted is at π and −π for different fractional delays, while the transition band of H-Farrow FB is centred at $\frac{\pi }{2}$ rad. The H-Farrow FB design is made up of three filtering stages; namely, the base filter, $H_{a}\left(z\right)$, the coefficient decimation filter, $H_{cd}\left(z\right)$ and the masking filter A(z). The design used the Parks–McClellan algorithm and the filter is realised using the direct transposed FIR in its implementation.

## 3. Hybrid Farrow Channelisation Algorithm

The procedure for the HFarrow channelisation algorithm is as follows:Normalise all the channel bandwidths (Cs), such that the $C_{i}$ and transition bandwidth $\Delta_{i}$ specifications range from 0 to 1; 1 corresponds to $\frac{f_{s}}{2}$, where $f_{s}$ is the sampling frequency.Compute the channel stop band frequency, $\omega_{si}$, such that $\omega_{si}=\frac{C_{i}}{2}$, where $C_{i}$ is the channel bandwidth.Determine the prototype stop band frequency as, $f_{proto}$=$\frac{GCD(C_{1}^{\prime },C_{2}^{\prime },C_{3}^{\prime })}{2}$ where $f_{proto}$ is the prototype stop band frequency and the procedure is to find the greatest common.The decimation factor *M* and the interpolation factor *L* of the filter are computed as follows, $M_{ma}=\frac{\pi }{\omega_{ms}}$, $L_{ma}=\left\lfloor \frac{\pi }{\omega_{ms}}\right\rfloor$. The value $d^{k}$ is computed using $\frac{L_{ma}}{M_{ma}}$, where $d^{k}$ is the fractional delay rate of the filter.The decimation factor $M_{mc}$ and the interpolation factor $L_{mc}$ for the complementary filter are computed as follows: $M=\frac{\pi }{\pi +\omega_{mcs}}$, $L=\left\lfloor \frac{\pi }{\pi +\omega_{mcs}}\right\rfloor$. Thus, the fractional rate for complementary filter can be calculated thus: $\frac{L_{mc}}{M_{mc}}$.Calculate the transition bandwidth for masking and complementary filter, using $\Delta_{k}^{\prime }=\Delta_{k}\times d^{k}$.Determine the base modal or complementary modal TBW as$\Delta_{modal}$= min $\left(\Delta_{1}^{\prime },\Delta_{2}^{\prime },\ldots ,\Delta_{n}^{\prime }\right)$. This corresponds to the modal transition width.Calculate the prototype, masking and complementary passband width using$\omega_{p}=\omega_{s}-\Delta_{k}^{\prime }$.Compute the stop band ripple and passband peak ripple using $\delta_{s1}^{\prime }=\delta_{s1}\frac{L_{i}}{M_{i}}$ and $\delta_{pmodal}$=min($\delta_{p1}^{\prime },\delta_{p2}^{\prime },\ldots ,\delta_{pn}^{\prime }$).Use the filter order $N=\frac{-2log_{10}\left(\delta_{p}^{\prime }\delta_{s}^{\prime }\right)}{3\Delta_{TBW}}$−1 [37] to calculate the channel filter length for prototype, masking and complementary filter.

Figure 5 represents the step-by-step procedure for demonstrating the HFarrow channelisation algorithm.

## 4. Simulations and Results

Using Matlab 2020 as the simulation tool, the developed hybrid Farrow algorithm was applied to Zigbee, Bluetooth (BT) and wideband code division multiplexers access (WCDMA). Channel bandwidth parameters used for BT, Zigbee, and WCDMA were 1 MHz, 4 MHz and 5 MHz, respectively. In addition, BT, Zigbee and WCDMA transition widths were specified as 50 kHz, 200 kHz and 500 kHz, respectively. The passband ripples and stop band attenuation for BT and Zigbee were specified as 0.1 and −40 dB, while WCDMA channel passband ripples and stop band attenuation were specified as 0.1 and −55 dB, respectively. The algorithm procedure in Section 3 was implemented and the filter results were recorded.

The metrics used for the computational complexity are filter order, number of multipliers, stop band attenuation and passband ripples.

Using the information contained in the algorithm procedure Section 3, the normalised channel bandwidth of BT, ZigBee and WCDMA were 0.05, 0.2 and 0.25, respectively. The channel bandwidths were 0.025, 0.1 and 0.125. From Step 2 of Section 3, the passband width of the prototype filter was set to the greatest common divisor of the signal bandwidth. The modal filter normalised frequency value was 0.025, which corresponds to the modal stop band frequency.

### 4.1. Performance of Modulation Farrow Algorithm

From Table 2, If k=2 and m=2 and N=40, the weighted scale is set to 0.000013 for BT, with stop band attenuation of 38.3, passband ripples of 0.09877 and filter order of 205. Zigbee has a filter order of 98 with stop band attenuation of 41.22 and passband ripples of 0.0989. When N=8, WCDMA signals have a filter order of 72 with the stop band attenuation of 56 and passband ripples of 0.997. The total number of multipliers used was 564. Also from Table 3, when k=2 and m=8 and N=40, the total number of multipliers used by the filter was 690.

### 4.2. Performance of Modulated Interpolated Coefficient Decimated Farrow Algorithm

Table 4 shows the frequency characteristics of masking filter using HFarrow algorithm. The following were the decimation factors for the modal filter, BT, Zigbee and WCDMA: $\frac{39}{40}$, $\frac{39}{40}$, $\frac{9}{10}$ and $\frac{7}{8}$, respectively. The modal decimation factor was found to be $\frac{39}{40}$. When a fractional rate, $d^{k}$, of $\frac{39}{40}$, was applied to the modal filter, the transition bandwidth computed was 0.002375, with a passband peak ripple of 0.1 dB, stop band peak ripple of −50 dB and the filter length of 132. In addition, when a fractional rate, $d^{k}$, of $\frac{39}{40}$, was applied to the BT channels, the transition bandwidth computed was 0,0026, with a passband peak ripple of 0.00975 dB, stop-band peak ripple of −39 dB, and a filter length of 107.

When the fractional rate of $\frac{9}{10}$ was applied to Zigbee, the transition bandwidth was calculated to be 0.011, with a passband ripple of 0.09, a stop band peak ripple of −39 and a filter order of 24. When the fractional rate of $\frac{7}{8}$ was applied to WCDMA, the transition bandwidth was calculated to be 0,021, with a passband ripple of 0.0875, a stop band peak ripple of −48.125 and a filter order of 9. In addition, the stop band for complementary masking frequency $\omega_{mcs}$ was calculated as shown in Table 5. The values of the stop band edge, passband edge and the fractional rate were calculated using design Steps 5 through to Steps 9 in Section 3. The complementary masking decimator factor for the modal filter, BT, Zigbee and WCDMA were $\frac{8}{9}$, $\frac{8}{9}$, $\frac{8}{9}$ and $\frac{7}{8}$, respectively. The complementary masking transition bandwidth for the modal filter, BT, Zigbee and WCDMA were 0.00222, 0.00222, 0.0089, 0.021875 with the filter order of 209, 150, 37 and 13.

Figure 6, Figure 7, Figure 8 and Figure 9 show the magnitude response of the modal filter, Bluetooth, Zigbee and WCDMA, respectively. Figure 6 shows the magnitude responses of the modal filter with stopband attenuation of −50 dB. Figure 7 shows the magnitude response of the BT masking filter when HFarrow operations were carried out with a fractional rate of $\frac{L}{M}$ equal to $\frac{39}{40}$, stop band attenuation of −39 dB and a filter order of 107. Figure 8 shows the magnitude responses of the Zigbee masking filter when HFarrow operations were carried out with fractional rate of $\frac{L}{M}$ equal to $\frac{9}{10}$, stop band attenuation of −39 dB and a filter order of 24. Figure 9 shows the magnitude responses of the WCDMA masking filter when HFarrow operations were carried out with a fractional rate of $\frac{L}{M}$ equal to $\frac{7}{8}$, stop band attenuation of −48.125 dB and a filter order of 9. The number of multipliers utilised by the HFarrow filter bank was analysed, compared and found to be lower than the CDFB [18] and ICDM [31] methods as indicated in Table 6 and Table 7.

From Table 7, the total number of multipliers used by the HFarrow filter bank was 389 while the ICDM expended 1545 multipliers, NU MDFFB consumed up to 1090 and CDFB used up to 1745. Thus, the total number of multipliers utilised by HFarrow channelisation was 22% of the total number of multipliers in the CDFB algorithm, while it depleted 25% of the total number of multipliers in ICDM. The percentage of multipliers used by the HFarrow algorithm in comparison with NU MDFT FB was 37%. HFarrow showed reductions in the following: 78% in comparison with CDFB, 75% in comparison with ICDM and 63% in comparison with NU MDFT. There was a remarkable reduction in the number of multipliers used in HFarrow compared with the algorithms used for the comparison as shown in the literature.

## 5. Conclusions

In this paper, a low-complexity Farrow channelisation algorithm based on a hybrid Farrow filter (HFarrow) method was designed by modulating the Farrow filter and cascading it with a frequency response masking interpolated coefficient decimating filter.

The investigation was carried out by test application of Bluetooth, Zigbee and wideband code division multiplexer access (WCDMA) to the HFarrow algorithm, by varying parameters such as filter order, stop band attenuation, passband ripples and the number of multipliers. The design example demonstrated that the HFarrow filter showed multiplier reduction rates as follows: 70% reduction in comparison with CDFB, 64% reduction rate in comparison with ICDM and 50% reduction in comparison with NU MDFT. Thus, the HFarrow filter should be a better choice of low-complexity multistandard receiver channelisation algorithm instead of the conventional Farrow method. However, this is at the expense of an increase in architectural design. In the future, a multiplierless hybrid Farrow filter should be considered. The main results obtained from HFarrow filter are as follows:The total number of multipliers used by the HFarrow filter bank were 389 while the ICDM expended 1545 multipliers, NU MDFFB consumed up to 1090 and CDFB used up to 1745.The total number of multipliers utilised by HFarrow channelisation was 22% of the total number of multipliers in the CDFB algorithm, while it depleted 25% of the total number of multipliers in ICDM.HFarrow showed reductions in the following: 78% in comparison with CDFB, 75% in comparison with ICDM and 63% in comparison with NU MDFT.

## Figures and Tables

**Figure 1 sensors-22-01164-f001:**
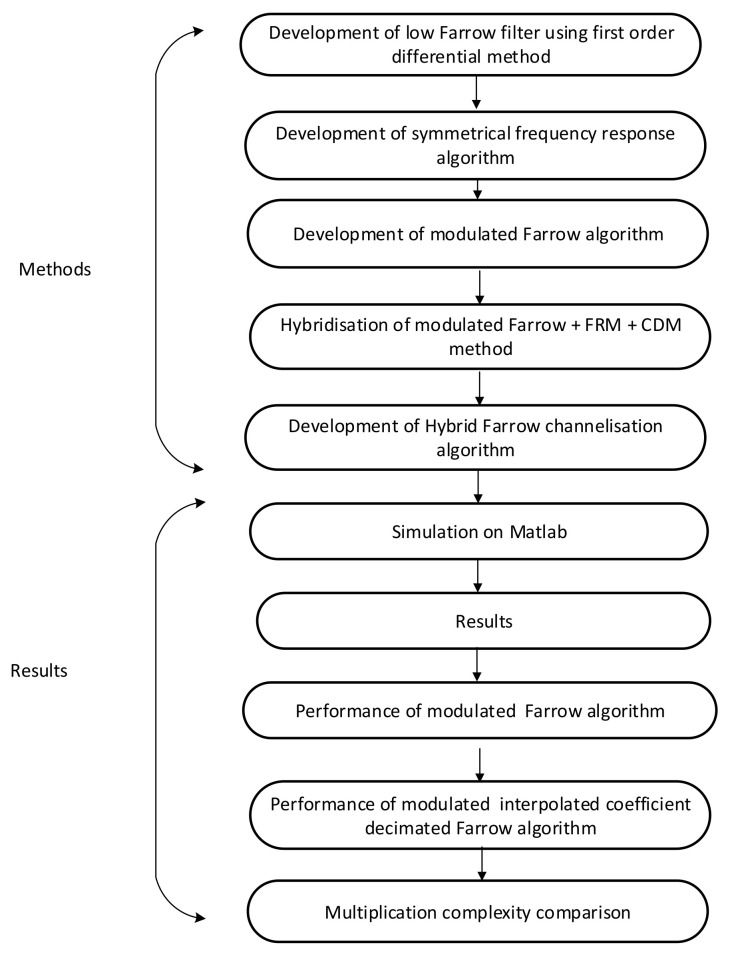
Flowchart depicting the HFarrow procedure.

**Figure 2 sensors-22-01164-f002:**
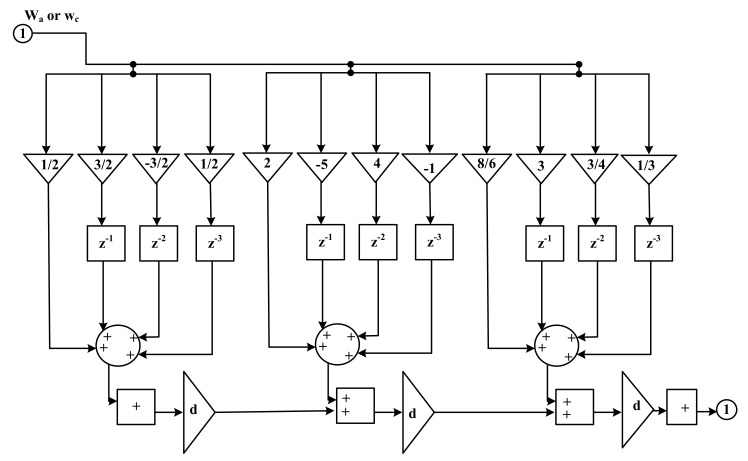
Farrow sub-filters.

**Figure 3 sensors-22-01164-f003:**
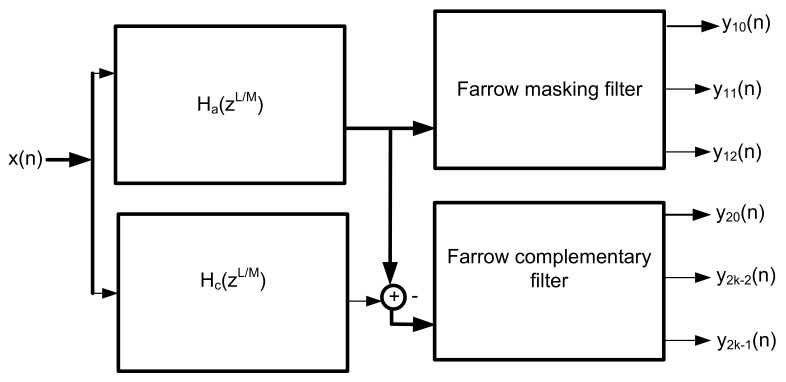
Block diagram of coefficient decimated FRM Farrow-based FIR filter.

**Figure 4 sensors-22-01164-f004:**
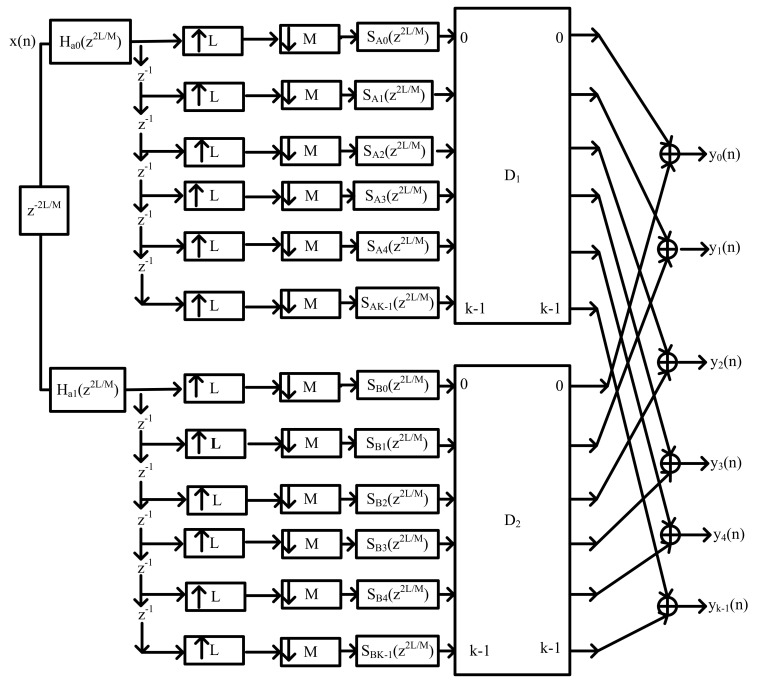
Diagram depicting HFarrow channelisation algorithm.

**Figure 5 sensors-22-01164-f005:**
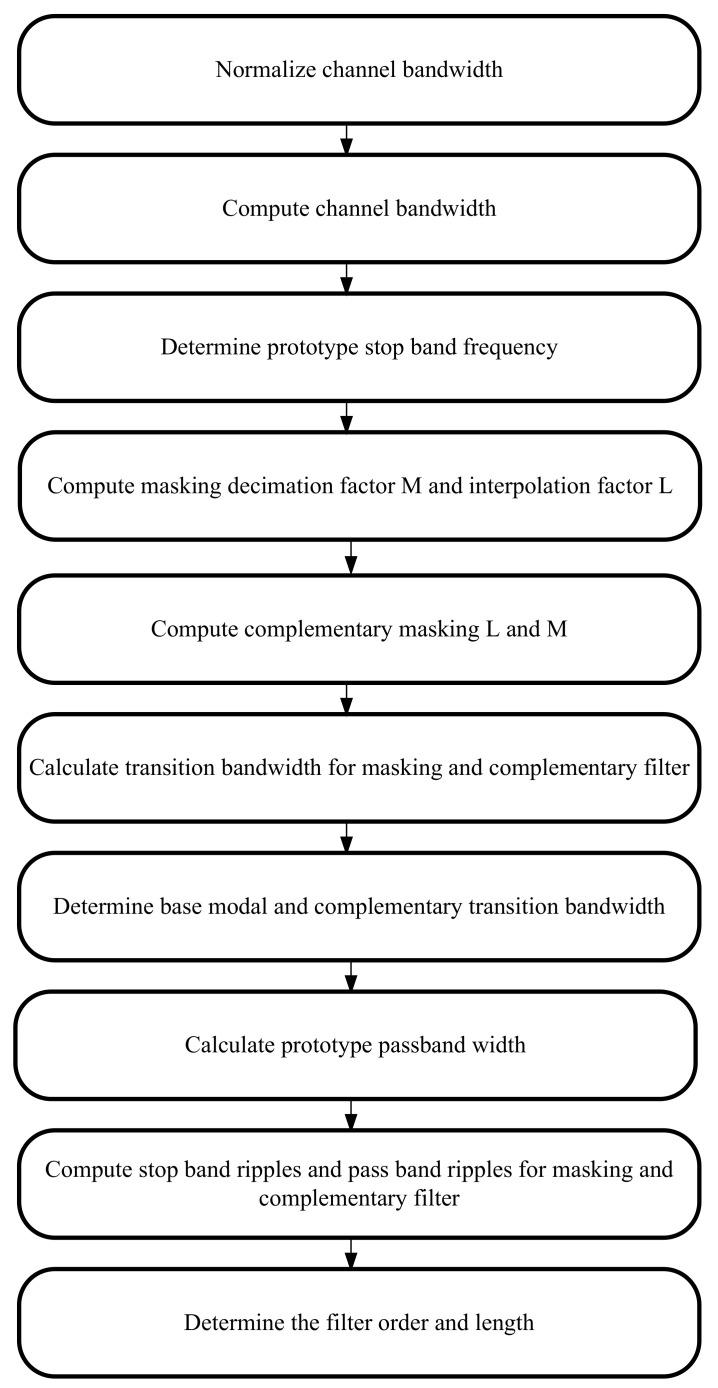
Step-by-step procedure for illustrating HFarrow filter.

**Figure 6 sensors-22-01164-f006:**
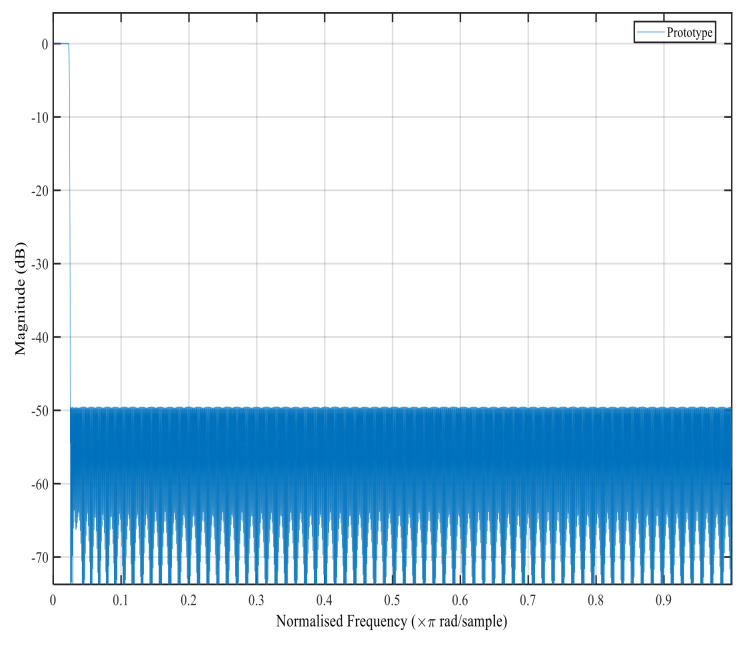
Magnitude response for the modal filter using the HFarrow algorithm.

**Figure 7 sensors-22-01164-f007:**
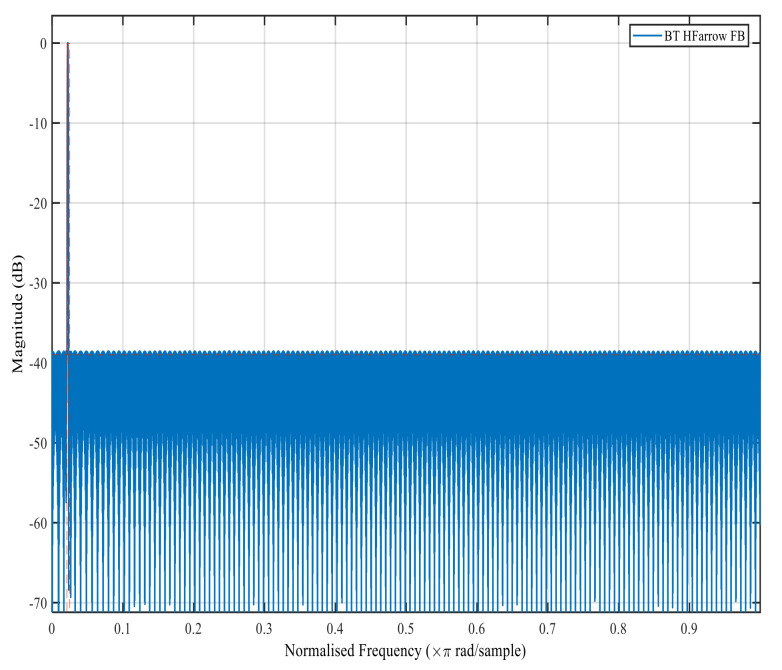
Magnitude response for the Bluetooth masking filter using the HFarrow algorithm.

**Figure 8 sensors-22-01164-f008:**
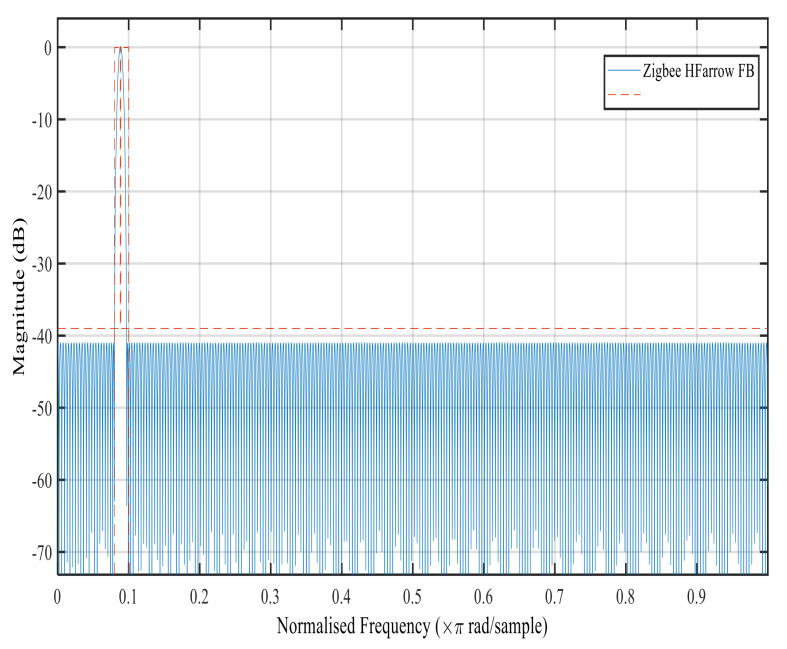
Magnitude response for the Zigbee masking filter using the HFarrow algorithm.

**Figure 9 sensors-22-01164-f009:**
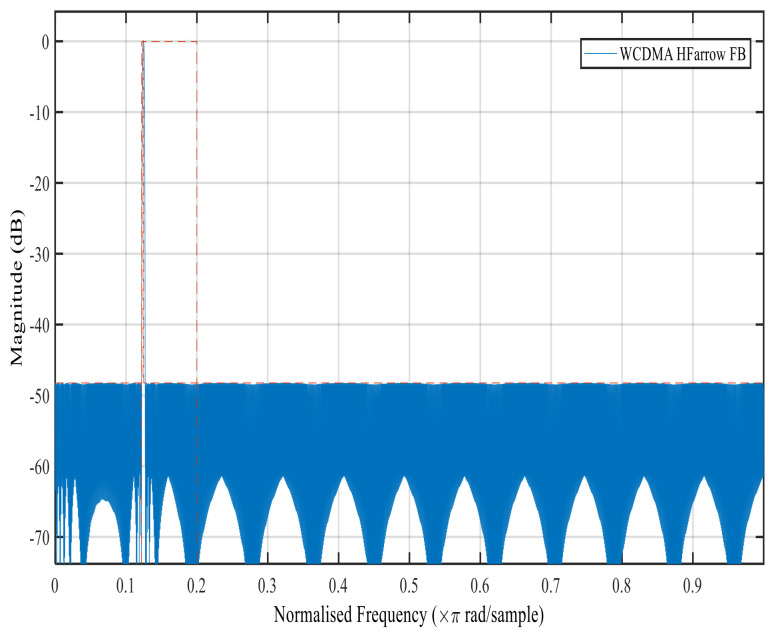
Magnitude response for the WCDMA masking filter using the HFarrow algorithm.

**Table 1 sensors-22-01164-t001:** Comparison study of the different channelisation algorithms.

Channelisation Algorithm	Computational Load
Modified Farrow [5]	Very High
Transposed modified Farrow [6,7]	High
Coefficient decimation type 1 [9,11,12,18]	Medium
Coefficient decimation type 11 [9,11,12,18]	High
MCD 1 and 11 [9,10,11,12]	Low
ICDM 1 and II [31]	Low
Interpolation FRM [20]	Very Low
CDM + interpolation FRM [20]	Low
ICDM [17,31]	Very High
Interpolation + Farrow structure [30]	Very High
Cosine modulated filter bank [3,32]	Low
FRM [15,31]	High
FRM on tree structure NUFB [9]	High
FRM NU MDFT [4,9]	High
HMICDM [29]	High

Very High: higher filter order and filter coefficients; High: high filter order and filter coefficients; Medium: medium filter order and filter coefficients; Low: low filter order and filter coefficients; Very Low: very low filter order and filter coefficients.

**Table 2 sensors-22-01164-t002:** The frequency characteristics of the modulated Farrow filter when m=2, k=2.

Filter Bank	*m*	*M*	Stop Band Frequency ($\omega_{ms}$)	Passband Frequency ($\omega_{mp}$)	Passband Ripples ($\delta_{ms}$)	Stop Band Attenuation ($\delta_{mp}$)	Filter Length
Modal filter, $H_{a}$	2	40	0.025	0.0225	0.9877	56.9	205
Bluetooth, $H_{ma}$	2	40	0.025	0.0225	0.998	43.9	189
Zigbee, $H_{ma}$	2	10	0.1	0.09	0.989	41.22	98
WCDMA, $H_{ma}$	2	8	0.2	0.175	0.997	56	72

**Table 3 sensors-22-01164-t003:** The frequency characteristics of the modulated filter when m=10.

Filter Bank	Stop Band Frequency ($\omega_{ms}$)	Passband Frequency ($\omega_{mp}$)	Passband Ripples ($\delta_{ms}$)	Stop Band Attenuation ($\delta_{mp}$)	Weight	Weight	Filter Length
Modal filter, $H_{a}$	0.025	0.0225	0.998	−58	10	39	240
Bluetooth, $H_{ma}$	0.025	0.0225	0.998	−58	10	39	240
Zigbee, $H_{ma}$	0.1	0.09	0.989	−62	10	39	120
WCDMA, $H_{ma}$	0.2	0.175	0.987	−68	10	670	90

**Table 4 sensors-22-01164-t004:** The frequency characteristics of masking filters implemented using the HFarrow filter bank.

Filter Bank	$d^{k}$	Stop band Frequency ($\omega_{ms}$)	Passband Frequency ($\omega_{mp}$)	Passband Ripples ($\delta_{ms}$)	Stop band Attenuation ($\delta_{mp}$)	Filter Length
Modal filter, $H_{a}$	$\frac{39}{40}$	0.025	0.022625	0.1	50	132
Bluetooth, $H_{ma}$	$\frac{39}{40}$	0.025	0.0224	0.0975	−39	107
Zigbee, $H_{ma}$	$\frac{9}{10}$	0.1	0.089	0.09	−39	24
WCDMA,$H_{ma}$	$\frac{7}{8}$	0.2	0.125	0.0875	−48.25	9

**Table 5 sensors-22-01164-t005:** The frequency characteristics of the complementary masking filter implemented using the HFarrow filter bank.

Filter Bank	$d^{kc}$	Stop band Frequency ($\omega_{ms}$)	Passband Frequency ($\omega_{mp}$)	Passband Ripples ($\delta_{mp}$)	Stop band Attenuation ($\delta_{ms}$)	Filter Length
Modal filter	$\frac{8}{9}$	0.027307	0.02269	0.1	−50	147
Bluetooth	$\frac{8}{9}$	0.027307	0.02269	0.092	−36.92	134
Zigbee	$\frac{8}{9}$	0.1080	0.0911	0.088	−35.5	29
WCDMA	$\frac{7}{8}$	0.2	0.125	0.0875	−48.25	9

**Table 6 sensors-22-01164-t006:** Multiplication complexity for non-uniform filter bank.

Filter Bank	$H_{a}$	Filter Order $H_{ma}$	$H_{mc}$	Total Number of Multiplications
Modal filter	279	-	-	187
BT	-	107	134	156
Zigbee	-	24	29	37
WCDMA	-	9	9	9

**Table 7 sensors-22-01164-t007:** Comparison of different multiplication complexities for non-uniform filter bank.

Filter Bank	$H_{a}$	Filter Order $H_{ma}$	$H_{mc}$	Total Number of Multiplications
CDFB [18]	3089	400	-	1745
ICDM FB [31]	2929	160	-	1545
NU-MDFT FB [4]	187	430	469	1090
HFarrow filter Bank	187	100	102	389

## Data Availability

Not applicable.

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
