# Peer review of "Low-Complexity Filter for Software-Defined Radio by Modulated Interpolated Coefficient Decimated Filter in a Hybrid Farrow"

_sensors, 2022, doi:10.3390/s22031164_

Round 1
Reviewer 1 Report
This paper proposes the hybrid farrow filter algorithm and its efficiency has been tested. This work is interesting and my observations are as follows:
1. There should be exhaustive comparison study of the different algorithm for the filter.
2. The contribution of the paper should be explicitly explained.
3. "Where" in line after (3) should be "where". Likewise after (34) and many other places.
4. In (5), D should be 'd'.
5. Fig. 1 and 2 should be explained.
6. Some perspective for near future improvements should be added.
Author Response
"Please see the attachment"

Reviewer 2 Report
This paper presents an improved version of Farrow filter. The improvement is resulting from hybridisation of Farrow filter with a specific masking filter. Though hibridisation is often used to achieve some better filtration quality (as even a typical e.g. band-pass filter can be considered as a hybrid solution comprasing of a low-pass and high-pass filters), then the solution presented in this paper is still of certain level of novelty due to an interesting combination of filters used. The resulting solution (HFarrow filter) proven to provide better filtration capability comparing to a range of other filters by allowing to extract higher frequencies and at the same time guarantee little distortion. It is also worth emphasising that despite visibly better performance the newly designed filter is not substantially more complex than the original filter so in practical implementation the improvements won't be computationally costly.
The Authors have provided a derivation of the new filter, designed a series of scenarios to test their solution and included enough results (gathered in a few tables) to prove that their solution indeed allows to achieve improved filtration quality.
The whole paper is written in good and comprehensible English and can be published as it is.
Author Response
"Please see the attachment"

Reviewer 3 Report
- You have to show the contributions of this study in some specific parts: (1) At the first part of the abstract (2) at the end of the introduction (3) at the first part of the conclusion.- You should follow this order for your revised abstract: (1) Novelty (2) Methods (3) Results (4) conclusion. Results are more important and you need to provide the most important outcomes here. Please revise the abstract accordingly. Please do not use abbreviations in the abstract. Also, avoid general sentences here.
- You have used a sum of several references in one reference i.e. lumping references. You need to avoid this. Separate the references and explain them separately.
- In the first part of the introduction, you need to discuss: why are researchers investigating and doing research on this topic? Why is this topic important?
- The literature review has been poorly written. In the literature review part, you should perform a potent literature review and scrutinize the most relevant and recent published papers in high-quality journal articles. You should check at least 30 relevant and up-to-date papers in this part. The literature review is one of the main parts of a scientific paper to show your novelty, and alert the readers that you are aware of the performed research studies.
- The last part of the introduction should conclude the limitations of the previous studies and provide the main objectives and novelties of this study. You need to clearly address the knowledge gap and provide some meaningful phrases that can advance the knowledge and can fill in a knowledge gap that has not been considered yet.
- You can use bullet points to show the main novelties of this work.
- You need a flowchart describing the whole investigation procedure to help the readers perceive the main points. In the flowchart, describe the methods used in this study step by step and also link the methods to the corresponding results.
- Describe the methods chronologically. This is very important to help the readers to replicate your results. Please cite previous research studies where necessary.
- The equations need some references. - The figures have not been appropriately explained as well. The readers cannot perceive the main points. Please describe the critical points and trends in the figures.
- You need to clearly describe the trends in the figures and justify them.
.- Compare your results with the literature so we can easily see your contributions.
- The conclusion should be majorly revised, and the main results should be summarized as the bullet points at the end of the conclusion. You should follow this structure: 1. A brief description of what you have done and what is your novelty? 2. How did you investigate the system and what are the main useful parameters? 3. Provide the primary results as the bullet points? 4. State the main limitations of this study and present some suggestions for future research.
Author Response
"Please see the attachment."
